# ADVERSARIALLY INJECTED DIAGNOSIS FOR COHERENT VISUAL AUTOREGRESSIVE GENERATION

## ABSTRACT

Visual Autoregressive (VAR) models, despite their formidable generative capabilities, accumulate local prediction errors across scales, leading to detail loss and local distortions. To address this, we introduce AID-VAR, a plug-and-play method that improves pretrained VARs via Adversarially Injected Diagnosis. Inspired by GANs, we train a discriminator to detect visual errors in generated samples and use an adversarial objective to pull generations toward the manifold of real images. To avoid the computational and stability issues of directly updating the VAR, we attach a lightweight guidance injector that conditions on previously generated scales of a pre-trained and frozen VAR and injects adversarial features to guide the next scale. To quantify reductions in cross-scale errors, we introduce the Inter-Scale Consistency Score (ISCS), which measures the fidelity of transitions between consecutive scales. Across standard VAR backbones, AID-VAR delivers sharper details, fewer local distortions, and stronger global coherence at remarkably low computational cost, adding negligible parameters and minimal computational overhead. Our results establish AID-VAR as a practical pathway for upgrading large VAR generators with adversarial feedback, without modifying training data, base architecture, or sampling schedules. For instance, our AID-VAR-d20 improves FID by 16%, with only 3% parameters increase.

## 1 INTRODUCTION

Autoregressive modeling has recently emerged as a competitive paradigm for high-fidelity image synthesis, offering stable training, exact likelihood, and flexible conditioning (Van den Oord et al., 2016b; Van Den Oord et al., 2017; Esser et al., 2021; Chang et al., 2022; Tian et al., 2024). In the visual autoregressive (VAR) (Tian et al., 2024) setting, an image is synthesized across a hierarchy of spatial scales; each stage conditions on previously generated content and predicts the next chunk of visual information. Despite their strong generative capacity, VARs suffer from a characteristic failure mode: local prediction errors accumulate across scales, compounding into detail loss, texture drift, and small-region distortions that undermine global coherence. These issues trace to exposure bias (Ranzato et al., 2015) and imperfect cross-scale transitions: once a local mistake enters the chain, subsequent decoders amplify rather than correct it.

We argue that mitigating these cross-scale errors requires diagnostic feedback during generation that can recognize "off-manifold" artifacts and nudge the process back toward natural images. Adversarial learning (Goodfellow et al., 2014; Brock et al., 2019; Karras et al., 2019; 2021) provides precisely such a signal: a discriminator trained on real *v.s.* generated images excels at detecting fine-grained visual pathologies (e.g., broken micro-geometry, aliasing, texture misalignment). However, naïvely adversarially fine-tuning the base VAR is often impractical: it introduces training instability, alters a large number of parameters, and demands substantial compute to retain the original model's strengths.

This paper introduces AID-VAR (Adversarially Injected Diagnosis for VAR), a lightweight, plug-and-play guidance module that upgrades a pre-trained, frozen VAR with adversarial feedback, without modifying training data, the base architecture, or the sampling schedule. AID-VAR attaches a small guidance injector to the VAR and trains a discriminator to detect visual errors. During generation, the injector conditions on previously generated scales and injects a spatial guidance signal that steers the next scale's prediction toward the natural-image manifold identified by the discrimi-

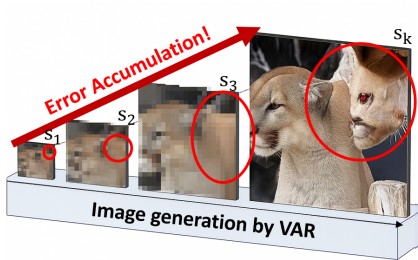 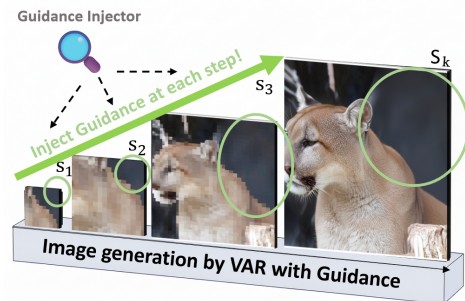

Figure 1: Conceptual illustration of our motivation. Left: The standard VAR model's coarse-to-fine generation is prone to error accumulation, where initial inaccuracies (circled in red) are magnified across scales, leading to severe structural degradation. Right: Our AID-VAR framework introduces a guidance injector at each step to anticipate and correct these errors, ensuring a globally coherent and plausible final image.

nator. Crucially, the base VAR remains frozen; only the tiny injector and the discriminator heads are trained, yielding stability, efficiency, and easy implementation across backbones.

We further propose two key strategies for an effective error fixing. i) Differentiable adversarial path in RGB-space. Training a discriminator in continuous latent spaces collapses quickly and provides weak signals. We instead operate in RGB with a projected discriminator (Sauer et al., 2021) that uses a fixed, pretrained image encoder to extract features and learns only shallow heads. To make the adversarial loss differentiable to the injector, we introduce soft-label based VQ-VAE decoding, which provides stable gradients from image space back to the injector. ii) Dynamically refreshed adversarial supervision. Rather than using a static pool of fakes that quickly becomes stale, we continuously refresh discriminator training with up-to-date generations from the current injector. Each iteration: (1) run the frozen VAR + injector to produce refined images via the soft-decoding path; (2) update the discriminator on real vs. current refined images; (3) update the injector using the discriminator's signal. This on-the-fly refresh reduces overfitting to outdated artifacts, stabilizes the min–max game, and improves sample efficiency.

Beyond a method, we also seek to measure what existing metrics under-capture: the quality of cross-scale transitions. Standard image-level scores (FID (Heusel et al., 2017), IS (Salimans et al., 2016), precision/recall (Kynkäänniemi et al., 2019)) reflect final realism but say little about whether each successive scale remains faithful to its coarse context. We therefore introduce the Inter-Scale Consistency Score (ISCS), which quantifies the fidelity of transitions between consecutive scales. ISCS serves as a targeted probe for exposure bias across the hierarchy: higher ISCS indicates that fine-scale refinements remain consistent with coarse-scale structure, aligning with perceptual coherence. Evaluated under ISCS, baseline VARs score low due to sensitivity to early errors and semantic drift; in contrast, AID-VAR substantially boosts ISCS, indicating stabilized cross-scale refinement and reduced error propagation.

Our contributions are four-fold:

- We propose a novel adversarial framework to solve the cumulated error issue in VAR, by diagnosing and correcting cross-scale errors via a lightweight injector.

- We make adversarial supervision effective and stable for VARs through an RGB-space discriminator and a differentiable soft-decode route that jointly deliver informative gradients to the injector, together with a dynamically refreshed training loop that maintains up-to-date fake samples.

- We introduce ISCS to quantify cross-scale coherence, complementing FID/IS by directly targeting exposure-bias effects between adjacent scales.

- We conduct experiments on ImageNet-1K(Deng et al., 2009), showing significant improvements over VAR. For instance, For instance, our AID- VAR-d20 improves FID by 16%, with only 3% parameters increments.

## 2 RELATED WORK

### 2.1 AUTOREGRESSIVE MODELS

The application of autoregressive (AR) models to image generation, inspired by their success in NLP (Brown et al., 2020; Radford et al., 2018; Devlin et al., 2019), has evolved through several paradigms. Early methods focused on next-pixel prediction on raster-scanned sequences (Van den Oord et al., 2016b;a). A major advance was the introduction of visual tokenizers like VQ-VAE (Van Den Oord et al., 2017) and VQGAN (Esser et al., 2021), which compress image patches into discrete tokens, shifting the task to next-token prediction (Ramesh et al., 2021; Yu et al., 2022; Gu et al., 2022) and significantly improving quality. While this approach achieved state-of-the-art results, it suffered from an efficiency bottleneck due to the large number of required decoding steps. The Visual Autoregressive (VAR) model (Tian et al., 2024) recently addressed this with a "next-scale prediction" paradigm, drastically reducing generation steps. Other related works on efficient autoregressive generation include non-sequential or parallel decoding methods (Chang et al., 2022; Li et al., 2023; Tschannen et al., 2023). Despite this progress, all AR frameworks are inherently susceptible to error accumulation due to their sequential nature. Our work, AID-VAR, directly tackles this persistent challenge by introducing an external guidance mechanism to correct the generative process without altering the base model.

### 2.2 ADVERSARIAL TRAINING FOR GENERATIVE MODEL ENHANCEMENT

Adversarial training, first introduced by Generative Adversarial Networks (GANs) (Goodfellow et al., 2014), has expanded beyond its original application of training standalone GANs (Radford et al., 2016; Karras et al., 2019; Brock et al., 2019; Zhang et al., 2019). A prominent trend in recent years has been the use of adversarial objectives to enhance and refine other classes of generative models, such as diffusion models (Dhariwal & Nichol, 2021; Ho et al., 2020; Song et al., 2021; Rombach et al., 2022). For instance, adversarial losses have been used to improve perceptual quality (Rombach et al., 2022), enable fast sampling (Lu et al., 2022; Sauer et al., 2023b), and distill pre-trained models (Sauer et al., 2024). Our training strategy is particularly inspired by Adversarial Diffusion Distillation (ADD) (Sauer et al., 2024), which effectively combines a distillation loss with an adversarial loss to achieve highly efficient, few-step image synthesis. AID-VAR specifically adopts the efficient discriminator architecture popularized by ADD and Projected GANs (Sauer et al., 2021): a powerful, pre-trained feature extractor is kept frozen, and only a lightweight classification head is trained on top of its features. However, our application of this principle is novel. Instead of integrating the adversarial loss into the training of the primary generative model, we leverage this highly efficient adversarial framework to exclusively train an independent, lightweight guidance module—the Guidance injector. This allows us to surgically enhance a pre-existing, frozen model without incurring the costs of end-to-end adversarial training.

## 3 AID-VAR FOR ERROR DIAGNOSIS

Given a frozen visual autoregressive model (VAR) that predicts multi-scale tokens, our goal is to reduce cross-scale error accumulation without touching the base model's weights, data, or sampling schedule. We introduce a lightweight *guidance injector* trained adversarially with a projected RGB-space *discriminator*. At each scale, the injector produces a spatial guidance map conditioned on previously generated content and injects it into the next-scale prediction. The overall framework of AID-VAR is illustrated in Figure 2.

### 3.1 LEARNING TO DISCRIMINATE SCALES

**Preliminary: next-scale prediction in VARs.** Visual autoregressive models synthesize an image over a coarse-to-fine hierarchy of scales. At scale $k$, the model predicts the next block of visual tokens conditioned on all previously generated tokens (and any optional external condition $c$, *e.g.*, class/text). Let $S_k$ denote the predicted tokens at scale $k$, $x_k$ the corresponding hidden state, and $z_k$ the pre-softmax logits. A generic VAR defines

$$z_k = f_{\text{VAR}}(x_k|\{S_{<k}\}), \quad S_k = \arg\max \text{softmax}(z_k), \tag{1}$$

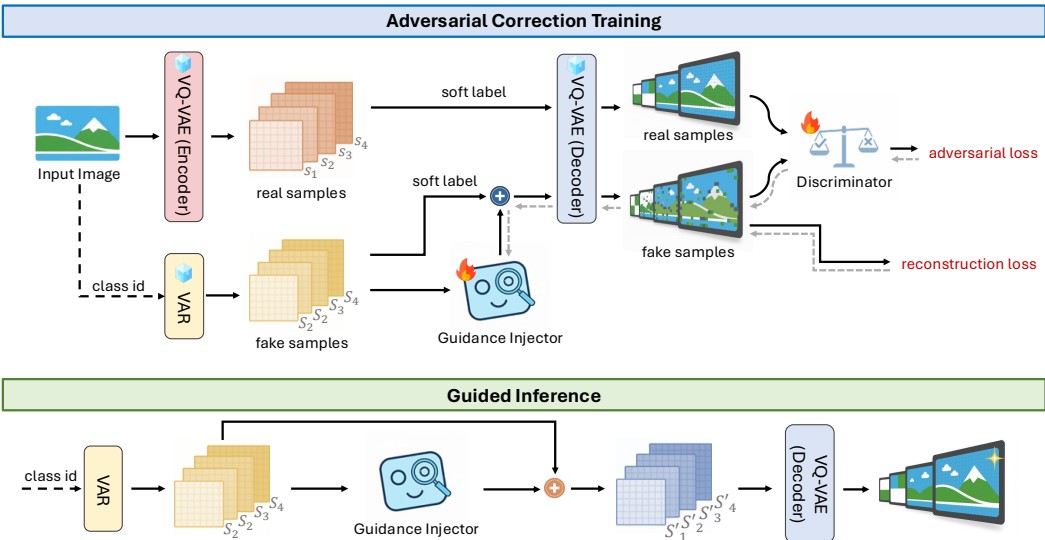

Figure 2: The AID-VAR framework for guided autoregressive generation. (1) Training: We freeze the pretrained VAR and learn a discriminator to adversarially train a lightweight guidance injector, which produces spatial guidance maps that are injected into the VAR's next-scale predictions via a differentiable soft-decoding path. (2) Inference: Only the guidance injector is attached to the frozen VAR to provide supplemental guidance signals, yielding a simple, efficient drop-in upgrade of VAR.

where each $x_k$ is conditioned on $\{S_{<k}\}$ and $c$. The process proceeds from coarse to fine and generates $K$ intermediate scales; after the finest scale, tokens are decoded to an RGB image by a VQ-VAE decoder (Van Den Oord et al., 2017).

**Learning to discriminate intermediate scale errors.** We seek a discriminator that is maximally sensitive to fine-grained visual defects that emerge in the generated intermediate scales, yet stable enough to supervise a lightweight guidance injector without updating the frozen VAR.

Therefore, following ADD (Sauer et al., 2024), a more promising way to construct the discriminator is leveraging a pre-trained visual foundation model (*e.g.*, DINO (Caron et al., 2021), CLIP (Radford et al., 2021)) that generalize well to the image samples. Formally, as the architecture shown in Figure 3(b), given an input image $I$, a frozen, pretrained vision encoder $E$ extracts features, which feed into shallow, learnable heads $h$:

$$D(I; \theta) := h(E(I); \theta), \tag{2}$$

where only $\theta$ is trained, and $E$ is frozen.

We empirically find this DINO-based discriminator enjoys multiple benefits compared to randomly-initialized discriminator:

- Perceptual alignment: pretrained encoders in RGB capture natural-image statistics that are directly predictive of aliasing, texture drift, and micro-geometry breaks.
- Stability: discriminating in discrete/token spaces introduces distributional mismatch and early collapse, whereas RGB features are smoother and more linearly separable for real/fake cues.
- plug-and-play: RGB-space supervision is agnostic to tokenizer/codebook design and generalizes across VAR backbones.

**Training data.** The discriminator itself can be regarded as a binary classifier to identify real and fake samples. So the training data of discriminator is composed with real samples and fake (generated) samples: (1) Real set: We sample the image $I$ uniformly from the training dataset of VAR, then feeds it into the VAR encoder to obtain tokens at each scale $k$, and use the decoder to obtain the corresponding images $I_k$ at each scale. (2) Fake set: we run the inference of the frozen VAR to generate next-scale tokens $S_k$ at each scale $k$. We then obtain the RGB images $\hat{I}_k$ via the decoder.

Figure 3: Architectures for the (a) Guidance Injector and (b) Discriminator. The injector is a lightweight Transformer encoder that processes previous scale features. The discriminator uses a frozen DINO backbone with a trainable classification head.

**Objective.** We use hinge loss (Lim & Ye, 2017) to train the discriminator on real and fake samples:

$$\mathcal{L}_D = \frac{1}{K} \sum_{k=1}^{K} \mathbb{E}_{I_k}[\max(0, 1 - D(I_k; \theta))] + \mathbb{E}_{\hat{I}_k}[\max(0, 1 + D(\hat{I}_k; \theta))]. \tag{3}$$

As a result, we now have a discriminator that can identify real images and fake generated images at each scale. Next, we will discuss how we use the discriminator to adversarially force the VAR to precisely generate images that lie in the manifold of real images.

## 3.2 ADVERSARIAL NEXT-SCALE PREDICTION WITH GUIDANCE INJECTOR

With a discriminator, the traditional adversarial approach updates the generator to fool a discriminator. Applied to VARs, this entails backpropagating through a large backbone and modifying millions (or billions) of weights, which (1) incurs substantial computational overhead; (2) destabilizes training under a min–max objective; and (3) risks irrecoverable drift in pretrained knowledge (catastrophic forgetting, degraded generalization). To avoid these pitfalls, AID-VAR is implemented as a lightweight extension to a frozen VAR: we keep **all** VAR parameters unchanged and learn only a small guidance module that nudges the next-scale prediction.

**Architecture of the guidance injector.** The guidance injector, $P_\phi$, is designed as a lightweight, spatially-aware module, as illustrated in Figure 3(a). It takes the feature map from the previously generated scale as input. The architecture consists of an input projection layer, an upsampling block to match the spatial dimensions of the next scale, followed by the addition of positional encodings. The core of the module is a series of two standard Transformer blocks (Vaswani et al., 2017), which process the spatially-rich features. Finally, an output projection layer produces the guidance feature map $G_k$, which has the same dimensions as the VAR's internal hidden state for the current scale. The predicted logits $z_k$ of each scale thus becomes:

$$z_k' = f_{\text{VAR}}(x_k + G_k | \{S_{<k}\}). \tag{4}$$

**Soft-label decoding for differentiability.** The VQ-VAE path in typical VARs is discrete, in which the direct token selection $S_k = \arg\max \text{softmax}(z_k)$ is non-differentiable. To optimize $P_\phi$ with adversarial signals in RGB space, we introduce a soft-label decoding route that preserves gradients from image space back to $z_k'$. The new decoding replaces the original discrete codebook lookup operation $H = W_{[S_k]}$ with

$$H' = W \cdot \text{softmax}(z_k'), \tag{5}$$

where $W$ is the codebook embedding matrix. This soft reconstruction enables a discriminator to act in RGB while providing stable gradients to the injector.

Note that to prevent the discriminator from exploiting reconstruction artifacts specific to the decoding route, we match its input pipeline for both real and generated samples in its training data. This means that both real and generated samples use soft-label decoding to obtain the images fed into the discriminator.

**Adversarial loss.** Similar to the adversarial loss in Equation (3), the injector at each scale $k$ is guided with a hinge-style adversarial loss:

$$\mathcal{L}_{\text{adv}} = -\mathbb{E}_{\hat{I}'_k}[D(\hat{I}'; \theta)] \tag{6}$$

, where $\hat{I}'_k$ is the predicted image with guidance injector.

To prevent over-aggressive deviation from the VAR prior and stabilize training, we add the token-level reconstruction loss $\mathcal{L}_{\text{rec}}$ in VAR as an auxiliary objective. The overall training objective for the guidance injector $P_\phi$ thus becomes:

$$\mathcal{L}_P = \mathcal{L}_{\text{adv}} + \lambda_{\text{rec}}\mathcal{L}_{\text{rec}}, \quad \text{with } \mathcal{L}_{\text{rec}} = \mathbb{E}_k[\text{CrossEntropy}(z'_k, s_k)]. \tag{7}$$

$\mathcal{L}_{\text{rec}}$ measures the cross-entropy between the guided logits $z'_k$ and ground-truth tokens $s_k$, and $\lambda_{\text{rec}}$ is a balancing hyperparameter.

### 3.3 On-the-Fly Discriminator Refresh for Up-to-Date Adversarial Signals

A key challenge in adversarial training is preventing the discriminator from overfitting to a fixed set of generated samples. If the discriminator is trained on a static dataset of "fakes", it quickly learns to identify the specific artifacts of the generator at that early stage. As the generator (in our case, the guidance injector) improves, the discriminator's feedback becomes outdated or "stale", providing a vanishingly small gradient and halting further improvement.

To circumvent this, we employ a dynamic, on-the-fly training strategy where the discriminator is continuously updated with the most recent outputs from the guidance injector. This ensures the adversarial signal remains relevant and challenging throughout the training process. The training loop for each batch proceeds as follows:

- **Generate current fakes:** For a given batch of input data, we first run the frozen VAR model augmented with the current guidance injector ($P_\phi$) to generate a set of fake images, $\hat{I}$, via the differentiable soft-decoding path.

- **Update discriminator:** Using the batch of real images $I_k$ and the newly generated fake images $I'_k$, we perform a single optimization step on the discriminator $D(\cdot; \theta)$ using the hinge loss ($\mathcal{L}_D$). This step teaches the discriminator to identify the specific flaws produced by the *current* version of the injector.

- **Update injector:** With the newly updated discriminator, we then perform a single optimization step on the guidance injector ($P_\phi$) using the combined adversarial and reconstruction loss ($\mathcal{L}_P$). The gradient from the discriminator provides a targeted signal that pushes the injector to fix its most recently identified failure modes.

This alternating update scheme creates a stable min-max game where both the injector and discriminator co-evolve. By constantly refreshing the training data for the discriminator, we prevent it from becoming stale and ensure that the guidance injector receives a consistent, high-quality learning signal, which is crucial for effectively minimizing error accumulation in the VAR.

### 3.4 Guided Inference

AID-VAR's inference follows a guided autoregressive procedure. The initial scale ($k = 1$) is generated using standard VAR sampling with Classifier-Free Guidance (CFG) (Ho & Salimans, 2022). For all subsequent scales ($k > 1$), the Guidance injector dynamically generates a guidance map $G_k$ from the previously sampled tokens $S_{k-1}$. This map is then injected into the VAR's internal state with a small, fixed weight before the current scale's tokens are sampled. This iterative, guided process continues until the final scale, after which the VQVAE decodes the complete token sequence into the final image.

## 4 Inter-Scale Consistency Score (ISCS)

The phenomenon of error accumulation in VAR models can be characterized as a scale-by-scale exposure bias (Ranzato et al., 2015; Bengio et al., 2015), where the model is conditioned on its own,

potentially flawed, outputs during inference. To specifically quantify this, we introduce the Inter-Scale Consistency Score (ISCS), a novel metric designed to measure the fidelity of the generative transitions between consecutive scales. Our core hypothesis is that a model's ability to progressively refine details and correct artifacts at finer scales is a critical indicator of its overall quality. Existing metrics like FID (Heusel et al., 2017) do not isolate this crucial aspect of refinement capability across the generative hierarchy.

To address this, we propose ISCS-FJD, which measures the Fréchet distance between the joint distributions of adjacent scales. For a large set of real and generated images, we construct two corresponding sets of joint feature vectors for each scale transition $k - 1 \rightarrow k$. Using a pre-trained DINO model (Caron et al., 2021) as a feature extractor, $\text{feat}(\cdot)$, we define the real joint distribution features $J_{\text{real}}^k$ and the generated joint distribution features $J_{\text{gen}}^k$ as:

$$J_{\text{real}}^k = \{[\text{feat}(s_{k-1}), \text{feat}(s_k)]\} \quad \text{and} \quad J_{\text{gen}}^k = \{[\text{feat}(r'_{k-1}), \text{feat}(r'_k)]\} \tag{8}$$

The per-scale score is the inverse of the Fréchet distance (FD) between these two sets of features:

$$\text{ISCS}_k = \frac{1}{\text{FD}(J_{\text{real}}^k, J_{\text{gen}}^k)} \tag{9}$$

A higher score indicates a smaller distance and thus better consistency. The final score is a weighted sum of the per-scale scores:

$$\text{ISCS} = \sum_{k=1}^{K} w_k \cdot \text{ISCS}_k \tag{10}$$

To emphasize the final stages of generation, we employ an exponential weighting scheme, $w_k \propto 2^k$, which assigns greater importance to finer scales (larger $k$). This ensures that the metric primarily rewards models that excel at high-fidelity detail synthesis and artifact removal during the final refinement steps.

## 5 EXPERIMENTS

### 5.1 EXPERIMENTAL SETTINGS

We conduct our training and evaluation on the ImageNet-1K dataset (Deng et al., 2009). All images are processed at a resolution of 256x256. The training is performed on the official training split, and all results are reported on the validation split.

**Evaluation metrics.** We evaluate the performance of our method using standard image generation metrics: Fréchet Inception Distance (FID) (Heusel et al., 2017), Inception Score (IS) (Salimans et al., 2016), Precision, and Recall (Kynkäänniemi et al., 2019; Sajjadi et al., 2018).

**Implementation details.** Our experiments are built upon three pre-trained and frozen VAR models of varying sizes: VAR-d16 (310M parameters), VAR-d20 (600M), and VAR-d24 (1.0B) (Tian et al., 2024). The trainable Guidance Injector is a lightweight Transformer with 2 layers and 8 attention heads. The discriminator is based on the StyleGAN-T architecture (Sauer et al., 2023a), using a frozen DINO ViT (Caron et al., 2021) as its feature extraction backbone.The Guidance Injector and the discriminator are both trained with a learning rate of 1e-6. For the composite loss of the Guidance Injector, the reconstruction loss weight is set to $\lambda_{\text{rec}} = 0.01$. The guidance weight during the training phase is fixed at $w = 0.001$.

### 5.2 PERFORMANCE ANALYSIS

We first present a comprehensive performance analysis of our proposed AID-VAR framework. We evaluate our method on the ImageNet 256x256 validation set and compare it against two primary sets of baselines: (1) the original, unguided VAR models of corresponding sizes, to directly measure the improvements conferred by our plug-and-play module, and (2) a wide array of state-of-the-art generative models, including leading GANs, diffusion models, and other autoregressive architectures.

The quantitative results are summarized in Table 1. Our findings clearly demonstrate the efficacy of the AID-VAR framework. For each model size (d16, d20, d24), our guided approach consistently

Table 1: Quantitative comparison on the ImageNet 256x256 validation set. Our AID-VAR models are compared against their unguided baselines and state-of-the-art generative models. '↓' indicates lower is better, while '↑' indicates higher is better.

| Type | Model | #Params | #Steps↓ | FID↓ | IS↑ | Pre↑ | Rec↑ |
|------|-------|---------|---------|------|-----|------|------|
| GAN | BigGAN (Brock et al., 2019) | 112M | 1 | 6.95 | 224.5 | 0.89 | 0.38 |
| | GigaGAN (Kang et al., 2023) | 569M | 1 | 3.45 | 225.5 | 0.84 | 0.61 |
| | StyleGAN-XL (Sauer et al., 2022) | 166M | 1 | 2.30 | 265.1 | 0.78 | 0.53 |
| Diff. | ADM (Dhariwal & Nichol, 2021) | 554M | 250+ | 10.94 | 101.0 | 0.69 | 0.63 |
| | LDM-4 (Rombach et al., 2022) | 400M | 250+ | 10.56 | 103.5 | 0.71 | 0.62 |
| | DiT-XL/2 (Peebles & Xie, 2023) | 675M | 250 | 9.62 | 121.5 | 0.67 | 0.67 |
| | L-DiT-7B (Alpha-VLLM, 2024) | 7B | 250 | 5.06 | 153.3 | 0.70 | 0.68 |
| | VDM++ (Kingma & Gao, 2023) | 2B | - | 2.40 | 225.3 | - | - |
| AR | VQGAN (Esser et al., 2021) | 1.4B | 256 | 15.78 | 78.3 | - | - |
| | RQ-Transformer (Lee et al., 2022) | 3.8B | 68 | 7.55 | 134.0 | - | - |
| | MaskGIT (Chang et al., 2022) | 227M | 8 | 6.18 | 182.1 | 0.80 | 0.51 |
| | MAGE (Li et al., 2023) | 230M | - | 6.93 | 195.8 | - | - |
| | MAGVIT-v2 (Yu et al., 2023) | 307M | - | 3.65 | 200.5 | - | - |
| | GIVT (Tschannen et al., 2023) | 304M | - | 5.67 | - | 0.75 | 0.59 |
| | MAR (Li et al., 2024) | 943M | - | 2.35 | 227.8 | 0.79 | 0.62 |
| | VAR-d16 (Tian et al., 2024) | 310M | 10 | 3.55 | 274.4 | 0.84 | 0.51 |
| | AID-VAR-d16 (Ours) | 321M | 10 | **3.24** | **280.0** | **0.85** | **0.51** |
| | VAR-d20 (Tian et al., 2024) | 600M | 10 | 2.95 | 302.6 | 0.83 | 0.56 |
| | AID-VAR-d20 (Ours) | 619M | 10 | **2.54** | **309.4** | **0.83** | **0.56** |
| | VAR-d24 (Tian et al., 2024) | 1.0B | 10 | 2.33 | 312.9 | 0.82 | 0.59 |
| | AID-VAR-d24 (Ours) | 1.02B | 10 | **2.08** | **316.3** | **0.82** | **0.60** |

outperforms its unguided VAR counterpart, achieving a notable reduction in FID while maintaining or improving other metrics. For instance, AID-VAR-d16 improves the FID score from 3.55 to 3.24. This enhancement is achieved with a negligible increase in parameters (approx. 11M), underscoring the efficiency of the Guidance Injector. When compared to the broader landscape of generative models, AID-VAR demonstrates highly competitive performance against top-tier diffusion models like MAR and VDM++, validating our approach as a practical and effective method for post-hoc enhancement of large-scale generative models.

## 5.3 Correction and Guidance Analysis

To qualitatively assess the impact of our framework, we conduct a visual analysis comparing the outputs of our AID-VAR with the unguided baseline VAR. The results, presented in Figure 4, reveal a consistent pattern: the baseline VAR model is prone to structural and anatomical errors due to error accumulation, while AID-VAR effectively corrects these flaws, leading to a significant improvement in global coherence and semantic plausibility.

A scale-by-scale analysis, presented in Table 2, validates our hypothesis that AID-VAR's primary strength lies in its late-stage refinement capabilities. While performance is comparable through the initial and mid-scales, a clear divergence emerges in the final high-resolution stages (scales 7-9). In this critical phase, AID-VAR's scores dramatically increase while the baseline stagnates, demonstrating a superior capability for detail synthesis and artifact correction. This decisive late-stage advantage, amplified by our weighting scheme, confirms the guidance

Table 2: Inter-Scale Consistency Score (ISCS) comparison. Higher scores indicate better consistency between generative scales. The final score is a weighted sum, prioritizing coarser scales.

| Scale | #Patch | VAR | AID-VAR |
|-------|--------|-----|---------|
| 0 | 1x1 | 0.48 | **0.52** |
| 1 | 2x2 | 0.97 | **1.53** |
| 2 | 3x3 | **2.58** | 2.55 |
| 3 | 4x4 | **4.64** | 4.24 |
| 4 | 5x5 | **7.68** | 5.91 |
| 5 | 6x6 | 8.63 | **10.52** |
| 6 | 8x8 | **14.95** | 14.86 |
| 7 | 10x10 | 20.04 | **22.37** |
| 8 | 13x13 | 25.14 | **38.17** |
| 9 | 16x16 | 25.32 | **45.87** |
| **Weighted ISCS** | | 22.64 | **38.80** |

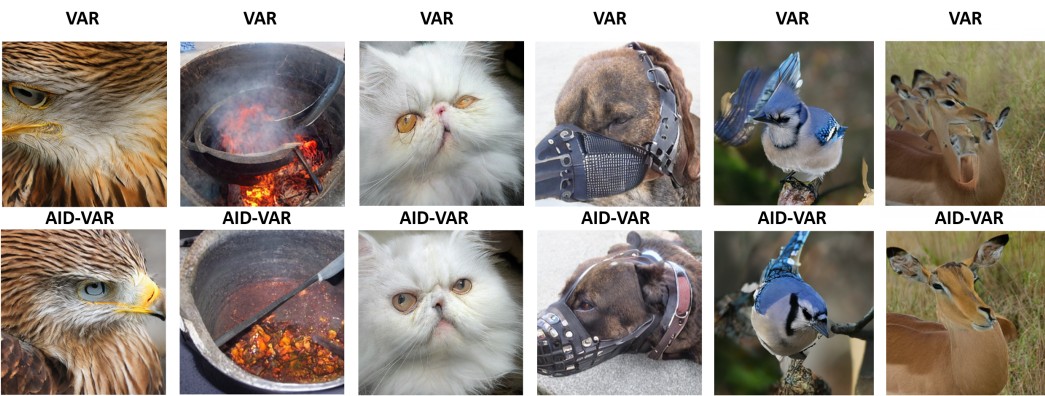

Figure 4: Qualitative comparison of AID-VAR against the baseline VAR. Our method consistently corrects a wide range of structural and semantic errors present in the unguided baseline.

injector acts as an expert finisher, elevating a competent draft into a high-quality final image, rather than correcting initial structural flaws.

The analysis can be stratified into three levels of improvement. First, AID-VAR demonstrates a powerful capability to fix **severe anatomical errors**. For instance, the baseline model generates distorted eyes for the kite and a catastrophic failure in rendering the Persian cat's eye. Our guided approach perfectly restores these key features with correct anatomy and realistic detail. Second, our method excels at improving **long-range structural coherence**. The baseline model erroneously produces a "ghost wing" on the blue jay, a clear structural fallacy. AID-VAR completely resolves this issue, generating a coherent and anatomically sound bird. Finally, AID-VAR enhances **subtle structural harmony**, as seen in the Siamese cat portrait, where it corrects unnatural facial swelling and inconsistent lighting present in the baseline generation.

## 5.4 COMPLEXITY ANALYSIS

A key advantage of the AID-VAR framework is its exceptional efficiency, enabling significant quality improvements with minimal resource overhead. The computational cost is negligible, as the lightweight Guidance Injector operates on the VAR model's pre-computed features and injects guidance via a simple element-wise addition. thus preserving the original model's high inference speed. Similarly, the parameter increase is exceedingly small, consistently remaining around 3% relative to the billion-parameter scale foundation models it enhances, adding only 11M, 19M, and 27M parameters for the VAR-d16, VAR-d20, and VAR-d24 models, respectively. This demonstrates that the AID-VAR framework leverages a remarkably small parameter budget and negligible computational cost to achieve significant gains in generation quality and global coherence, validating it as a highly efficient plug-and-play upgrade for large-scale visual autoregressive models.

## 6 CONCLUSION

To address the detail loss and structural distortions in Visual Autoregressive (VAR) models caused by error accumulation, we propose AID-VAR, a lightweight, plug-and-play framework. Without altering the pre-trained and frozen VAR model, our method introduces a small guidance injector that corrects the generative process via Adversarially Injected Diagnosis. To ensure stable and efficient training, we introduce two key strategies: a differentiable path from an RGB-space discriminator to the injector, enabled by soft-label decoding, and a dynamically refreshed adversarial supervision loop that provides a consistently effective training signal. Furthermore, we propose the novel Inter-Scale Consistency Score (ISCS) to specifically quantify the fidelity of transitions between adjacent scales, directly measuring the mitigation of error accumulation. Experiments demonstrate that AID-VAR significantly improves the performance of VAR backbones (for instance, a 16% FID improvement for AID-VAR-d20) with a minimal parameter increase (only 3%) and computational overhead, establishing a practical and efficient pathway for upgrading existing large-scale generative models.

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

## A  APPENDIX

This appendix provides additional details to supplement the main paper. Section B covers implementation specifics, including training setup and hyperparameters. Section C provides detailed model architectures.Section D presented more experiments and analysis. Section E discusses the limitations of our method and potential future work. Finally, Section F provides a broader impact statement.

## B  IMPLEMENTATION DETAILS

### B.1  TRAINING SETUP

All experiments were conducted on a server equipped with 8 NVIDIA A800 GPUs. The training of the Guidance Injector and discriminator for each VAR model variant on the full ImageNet dataset took approximately 10 hours. We use PyTorch for our implementation.

### B.2  OPTIMIZER AND HYPERPARAMETERS

We use the AdamW optimizer for training both the Guidance Injector and the discriminator heads. The optimizer parameters are set to $\beta_1 = 0.9$, $\beta_2 = 0.999$, and a weight decay of $1 \times 10^{-5}$. We use a constant learning rate throughout the training process, as we did not find significant benefits from using a learning rate scheduler with warmup or decay for our setup. Key hyperparameters for our main ImageNet training and the single-class training setup are detailed in Table 3.

## C  MODEL ARCHITECTURES

### C.1  GUIDANCE INJECTOR

The Guidance Injector is a lightweight Transformer encoder designed to be computationally efficient. It consists of 2 Transformer blocks with 8 attention heads each. The embedding dimension (`PLANNER_DIM`) is matched to the VAR model's depth, specifically `VAR_DEPTH * 64` (e.g., 1536 for VAR-d24). Within each block, the feed-forward network (FFN) has a hidden dimension of $\min(\text{embed\_dim} \times 2, 512)$ and uses the GELU activation function. This means the MLP ratio is approximately 0.5, 0.4, and 0.33 for our d16, d20, and d24 models, respectively. We employ a custom `SafeLayerNorm` for normalization and scale the residual connections by a factor of 0.1 to stabilize training. Sinusoidal positional encodings are added to the input features.

### C.2  DISCRIMINATOR

Our discriminator follows the Projected Discriminator architecture. We use a pre-trained DINO ViT-S/16 model (Caron et al., 2021) as the frozen feature backbone. The trainable part consists of shallow classification heads ("DiscHead"). Each head is composed of a main block and a classification layer. The main block includes a 1x1 convolution followed by a "ResidualBlock" containing a

Table 3: Key hyperparameters for full ImageNet training and single-class training.

| Parameter | Full ImageNet | Single-Class |
|---|---|---|
| VAR Depth | 16 | 16 |
| Epochs | 2 | 10 |
| Global Batch Size | 256 | 32 |
| LR (Guidance Injector) | $1 \times 10^{-6}$ | $5 \times 10^{-6}$ |
| LR (Discriminator) | $1 \times 10^{-6}$ | $1 \times 10^{-6}$ |
| Gradient Clip Value | 0.5 | Not specified |
| $\lambda_{\text{rec}}$ | 0.0 | 0.01 |
| Guidance Weight (train) | 0.001 | 0.001 |
| R1 Gamma | 0.2 | Not specified |

Table 4: Ablation studies on the ImageNet 256x256 validation set using VAR-d16 as the base model.

| Experiment | Configuration | FID↓ |
|---|---|---|
| Guidance Method | Baseline (No Guidance) | 3.55 |
| | Single Token Broadcast | 3.43 |
| | **Spatial Map (Ours)** | **3.24** |
| Discriminator Input | Baseline (No Guidance) | 3.55 |
| | VQ-VAE Features | 4.17 |
| | **RGB Image (Ours)** | **3.24** |
| Guidance Weight ($w$) | Baseline (w=0) | 3.55 |
| | 0.0001 | 3.56 |
| | **0.001 (Ours)** | **3.24** |
| | 0.01 | 4.32 |
| | 0.1 | 7.73 |

9x9 convolution. The classification layer is a 1x1 "SpectralConv1d" layer. During training, the entire DINO backbone remains frozen, and only the parameters of these heads are updated, ensuring efficiency and stability.

## D  ABLATION STUDY AND MORE ANALYSIS

We conduct a series of ablation studies to validate the key design choices of our AID-VAR framework. All ablations are performed on the VAR-d16 model.

**Guidance Method.** We first investigate the importance of spatial information in our guidance mechanism. We compare our standard approach, which uses a spatially-aware guidance map, against a simplified baseline where a single guidance vector is broadcast across all spatial locations. As shown in Table 4, while the single-vector approach (FID 3.43) offers a marginal improvement over the unguided baseline (FID 3.55), our spatial map (FID 3.24) yields a significantly larger gain. This result underscores that the efficacy of our method stems not from a global directional signal, but from its ability to apply spatially nuanced, targeted corrections, which is critical for resolving local structural errors.

**Discriminator Input Space.** Next, we validate our choice of using a pixel-space discriminator. We compare our method against an alternative where the discriminator operates directly on the VQ-VAE's intermediate features instead of decoded RGB images. The results in Table 4 are stark: training with features leads to a catastrophic failure (FID 4.17), performing substantially worse than even the unguided baseline. This confirms our hypothesis that a discriminator pre-trained on natural images (like the DINO backbone) is insensitive to the abstract, quantized latent space of the VAE. Operating in the RGB domain is therefore fundamental, ensuring the discriminator provides a meaningful adversarial signal for the Guidance Injector to learn from.

**Guidance Weight.** Finally, we examine the sensitivity of our model to the guidance weight, $w$, used during inference. We test a range of values, and the results in Table 4 reveal a distinct U-shaped performance curve. A weight that is too low ($w = 0.0001$) provides a negligible signal, resulting in performance almost identical to the baseline. Conversely, excessively high weights ($w \geq 0.01$) drastically degrade performance. This demonstrates that the guidance must act as a gentle correction rather than a forceful override of the base VAR model's powerful learned priors. Our chosen weight of $w = 0.001$ strikes the optimal balance, effectively correcting errors without disrupting the foundational generative capabilities of the frozen VAR model.

To visualize the underlying mechanism of the I-predictor, Figure 5 provides a step-by-step comparative analysis, revealing how our guidance prevents two critical failure modes of the unguided VAR model: catastrophic structural forgetting and subsequent structural hallucination. The visualization reveals a clear divergence in the early scales (1-4), where the baseline model (bottom row) collapses into an amorphous state, "forgetting" the subject's complex global structure in favor of a simpler

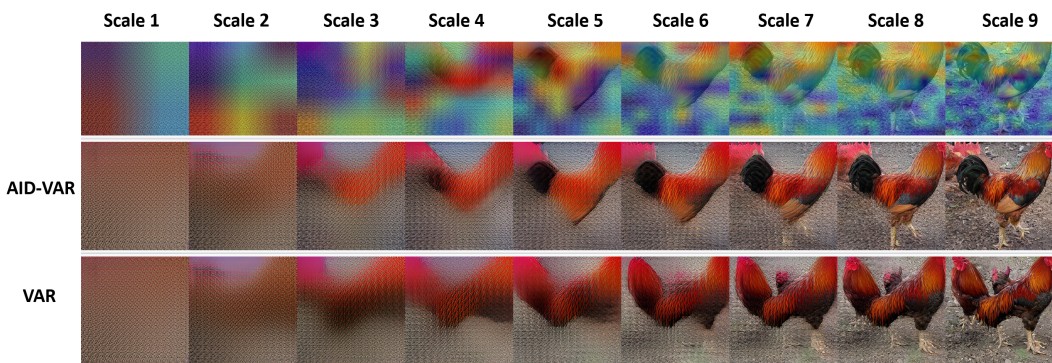

Figure 5: Visualization of AID-VAR as a Process Supervisor. This step-by-step comparative analysis reveals how our guidance prevents two critical failure modes of the unguided VAR model: catastrophic structural forgetting and subsequent structural hallucination. The visualization shows a clear divergence in the early scales (1-4), where the baseline model (bottom row) collapses into an amorphous state, "forgetting" the subject's complex global structure. In stark contrast, AID-VAR's trajectory (middle row), steered by a broad foundational guidance signal (top row), preserves the subject's structural integrity. This divergence becomes critical in the mid-level scales (5-7); having lost the correct structure, the baseline suffers from "structural hallucination" (e.g., rendering multiple heads), while the guidance signal intelligently focuses on these key regions to ensure AID-VAR materializes them correctly.

texture-filled mass. In stark contrast, AID-VAR's trajectory (middle row), steered by a broad foundational guidance signal (top row), preserves the subject's structural integrity from the outset. This initial divergence becomes critical in the mid-level scales (5-7); having lost the correct underlying structure, the baseline suffers from "structural hallucination," rendering features like multiple heads in anatomically nonsensical locations, while the guidance signal intelligently focuses on these key regions to ensure AID-VAR materializes them correctly. This culminates in two opposing outcomes: a catastrophic failure for the baseline versus a coherent, realistic image for AID-VAR. This analysis demonstrates that the I-predictor acts not merely as a refiner, but as a process supervisor, ensuring the entire generative sequence remains on a semantically and structurally plausible track by fundamentally preventing the step-by-step error accumulation that leads to generative failure.

## E  LIMITATIONS AND FUTURE WORK

Our work has several limitations that open avenues for future research. First, the effectiveness of the Guidance Injector can be sensitive to scenes with extremely high-frequency details, complex occlusions, or subjects from long-tail classes, where the simple additive guidance may not be sufficient to resolve all artifacts. Second, the guidance weight $w$ is a sensitive hyperparameter; while we found a value that works well across models, different datasets or resolutions might require re-tuning. The alternating training scheme, although stable, also introduces additional computational overhead compared to standard VAR inference.

Future work could explore making the guidance weight $w$ adaptive, allowing it to be dynamically adjusted based on the scale, content, or model's confidence. Another promising direction is to distill the knowledge from the discriminator directly into the Guidance Injector, a technique known as distillation, potentially removing the need for the discriminator during inference and reducing the computational cost. Finally, incorporating more robust learning rate strategies, such as the cosine annealing schedule available in our codebase, could further improve training stability and performance, especially for longer training runs or on larger-scale datasets.

## F BROADER IMPACT STATEMENT

**Positive Impacts:** The AID-VAR framework presents a practical and resource-efficient paradigm for improving existing large-scale generative models. This can have positive impacts in various domains. In creative fields, it can help artists and designers generate higher-quality, more coherent images, serving as a more reliable tool for concept art and automated content creation. In scientific applications, such as medical or satellite imaging, our method's ability to reduce structural artifacts could lead to more robust and reliable image reconstruction and enhancement, aiding in diagnosis and analysis. More broadly, our work contributes to the field of trustworthy AI by providing a method to diagnose and correct flaws in foundational models post-hoc.

**Potential Risks:** As with any powerful generative technology, there are potential risks. The ability to generate more realistic and coherent images could be misused for creating high-quality "deepfakes" or other forms of synthetic media for malicious purposes, such as spreading misinformation or creating fraudulent content. The increased quality might make such fakes harder to detect. Furthermore, while we aim to correct errors, the adversarial training process could inadvertently amplify existing biases present in the training data if not carefully monitored. The additional training step, though efficient, still contributes to the overall energy consumption and carbon footprint of developing large AI models.

**Mitigation Strategies:** To mitigate these risks, we support the concurrent development of robust detection methods for synthetic media. Techniques like digital watermarking can be integrated into the generative process. We advocate for responsible use policies and clear labeling of synthetic content when deployed in public-facing applications. To address bias, it is crucial to use diverse and carefully curated datasets and to continuously evaluate model outputs for fairness across different demographic groups. Finally, we encourage research into more energy-efficient training and optimization techniques to reduce the environmental impact.

## G ETHICS STATEMENT

This work does not raise any ethical concerns. All experiments were conducted on public datasets with appropriate licenses.

## H REPRODUCIBILITY STATEMENT

We will release all code to ensure reproducibility of our results.

## I USE OF LLMS

This work used a large language model (LLM) solely for minor language polishing. All ideas, methods, experiments, and analyses were entirely designed, implemented, and validated by the authors, who take full responsibility for the content of this paper.

