# OpenReview forum: "Adversarially Injected Diagnosis for Coherent Visual Autoregressive Generation"
_ICLR.cc/2026/Conference — ICLR 2026 Conference Withdrawn Submission_

### Official Review · Reviewer_zatQ · 2025-10-28

**Soundness:** 3
**Presentation:** 2
**Contribution:** 2
**Rating:** 4
**Confidence:** 5

**Summary:**

This paper proposes AID-VAR, a method to mitigate error accumulation in pre-trained Visual Autoregressive (VAR) models. The core idea is to attach a lightweight, trainable "guidance injector" to a frozen VAR. This injector is trained adversarially using a discriminator that operates in the RGB space. The injector conditions on features from previously generated scales and produces a spatial guidance map that is added to the VAR's internal state to steer the next-scale prediction. A "soft-label decoding" is designed to ensure differentiability from RGB back to the injector. The paper also introduces a new metric, the Inter-Scale Consistency Score (ISCS), to quantify the coherence between consecutive generation scales. Experiments on ImageNet demonstrate improvements in FID, IS, and the proposed ISCS over the base VAR models.

**Strengths:**

The idea of using a small, adversarially-trained add-on module to correct a large, frozen VAR model is novel and practical.

The paper provides a thorough quantitative evaluation against strong baselines.

The introduced ISCS metric may be useful in evaluating the multi-scale refinement process of VAR-like models.

**Weaknesses:**

The writing, while generally understandable, suffers from verbosity and a lack of precision in several critical sections. This reduces the paper's impact and makes it difficult to grasp the finer details of the method. Inconsistent and sometimes unclear notation further exacerbates this issue.

The central claim—that AID-VAR mitigates "local prediction errors accumulate across scales"—is not sufficiently analyzed or visualized. The qualitative results (Fig. 4) show final image improvements but do not clearly demonstrate the process of error correction across scales. Figure 5, intended for this purpose, is confusing and seems to show that AID-VAR also suffers from significant structural issues, which contradicts the narrative of preventing error accumulation from the start.

Several important implementation details are either missing from the method description or are confusingly presented. This lack of rigor undermines the reader's confidence and hampers reproducibility.

**Questions:**

The paper would be significantly strengthened by a more detailed analysis of the VAR's failure mode. How exactly do "local errors" manifest and propagate? Please provide clearer visualizations to demonstrate that AID-VAR specifically reduces error accumulation, rather than just improving the final output.

In Figure 5, the AID-VAR generation (middle row) at scale 9 appears to suffer from severe structural degradation around the tail. Please clarify this result.

The notation needs careful revision. There is no distinction between scalars, vectors and matrices. Important definitions, such as that of $f_{VAR}$, are missing. Please review all equations for consistency and clarity.

The guidance weight parameter w mentioned in Sec. 5.1 is not defined in the methodological sections. Please introduce it formally when the guided inference process is described.

In Figure 2, the terms "real samples" and "fake samples" are used to denote multiple variables, which is confusing. Furthermore, the real/fake samples of $S_1$, $S_2$, $S_3$ and $S_4$ may be confused with the discrete code after vector quantization. This is a critical detail that should be made explicit in the main text and in Figure 2.

What is the impact of using soft-label decoding on the real images fed to the discriminator? Does this introduce blur or artifacts that might weaken the adversarial signal? A brief discussion or analysis is needed.

What are the differences between the training loop in Sec. 3.3 and that of the standard GANs? The claim that it is a "key strategy" for "on-the-fly refresh" should be rephrased to clarify its novelty.

Why should we use the inverse Fréchet Distance in Equation (9), instead of the FD itself mimicking the FID?

---

### Official Review · Reviewer_BkNc · 2025-11-02

**Soundness:** 3
**Presentation:** 3
**Contribution:** 2
**Rating:** 6
**Confidence:** 3

**Summary:**

This work targets improving autoregressive image generation quality.
To do this, they employ adversarial training with a GAN inspired objective with a DINO-pretrained discrimnator.

Their architecture consists of:
- Base visual autoregressive model (VAR), which they keep frozen
- Lightweight guidance injector to refine intermediate features
- Discriminator head using frozen DINO ViT-S/16 backbone

For training, they use soft-label decoding (straight through estimator) to enable gradients to flow from the discriminator loss to their guidance injector.

They evaluate on ImageNet 256×256 using FID, IS, and their proposed ISCS (Inter-Scale Consistency Score) metric.
Results show FID improvements over their baseline VAR-d16/d20/d24 with negligible parameter increase.

**Strengths:**

The paper is well-written with clear presentation throughout.
The technical sections are detailed and easy to follow.

The results show good improvements in FID and IS across multiple VAR backbones.
A key strength is that these gains are achieved while keeping the VAR backbone frozen with minimal architectural changes.

**Weaknesses:**

The paper's experimental section is too concise and does not provide enough insight on some of the architecture/design choices involved in designing their method.
Table 4 from the appendix provides some analysis on the guidance methods, but this should be included in the main paper, along with more of the analysis.

Since the method relies on the discriminator training signal, ablations comparing the frozen VAR approach against alternatives (e.g., finetuning all weights or selectively unfreezing certain layers) would give insight on the effectiveness of the guidance injector.

**Questions:**

Figure 2 and the inference procedure (Section 3.3) state that "only the guidance injector is attached to the frozen VAR" during inference.
However, from the future work section in the appendix, there is a line "potentially remove the need for the discriminator during inference."
This statement is a bit puzzling - could the authors clarify whether the discriminator is currently used during inference?

There appears to be some discrepancy between the baseline performance numbers in Table 1 and those reported in the original VAR Table 1.
For example, L-DiT-7B is reported with FID 5.06 and IS 153.3 in Table 1, but VAR's table 1 reports L-DiT-7B with FID 2.28 and IS 316.2.
Could the authors clarify whether the evaluation protocol differs?

How important is the DINO discriminator to the method's success?
It would be helpful to see ablations comparing DINO against other pretrained visual encoders, or a discriminator trained from scratch.

---

### Official Review · Reviewer_dvdy · 2025-11-02

**Soundness:** 3
**Presentation:** 2
**Contribution:** 3
**Rating:** 2
**Confidence:** 4

**Summary:**

This work addresses a key limitation of existing VAR methods: the accumulation of local prediction errors across scales, which leads to a loss of detail and local distortions. To tackle this, the authors introduce a GAN-like discriminator that adversarially guides the generation process toward the real data manifold. They also propose a "consistency score" to quantify the fidelity of transitions between consecutive frames. Experimental results demonstrate that this approach achieves improved performance with only a minor increase in parameters.

**Strengths:**

The manuscript addresses the challenging task of rectifying local imperfections in image generations. The proposed method is novel and demonstrates a marked improvement in generation quality, as evidenced by the experimental results.

**Weaknesses:**

The manuscript's description of the methodology lacks clarity. Furthermore, the experimental design is inadequate, failing to validate the method's effectiveness from multiple perspectives.

**Questions:**

1. There is an apparent contradiction in the interpretation of the ISCS metric. An effective rectification of local errors by AID-VAR should logically lead to a lower ISCS, indicating more substantial corrective changes. This makes the reported higher ISCS values counter-intuitive and requires clarification.

2. The design choice of a single, shared discriminator for all scales warrants justification. The motivation is unclear, and the potential benefits of scale-specific discriminators should be discussed.

3. The description of "on-the-fly discriminator refresh" in Section 3.3 is confusing and frames a standard GAN training procedure as a novel component. It is standard practice to update the discriminator concurrently with the generator.

4. The validity of ISCS as a measure of inter-scale consistency is questionable. The calculation in Eq. (9) operates within individual scales and does not directly compare features between scale k-1 and k.

5. To fully showcase the model's capabilities, I suggest expanding the evaluation to include reconstruction quality, latent space controllability, and interpolation smoothness.

---

### Official Review · Reviewer_B48p · 2025-11-04

**Soundness:** 1
**Presentation:** 3
**Contribution:** 2
**Rating:** 0
**Confidence:** 4

**Summary:**

In this paper, the authors present AID-VAR which is a small auxiliary network trained to inject corrections to the VAR model's visual tokens at a given scale to improve the generation quality. The authors claim that this improvement in the generation quality is due to artifacts formed in the VAR model in earlier scales that propagates with prominence to higher scales . The paper presents results that show an improvement of FID of 16% relative for a 3% param increase without training/fine-tuning the VAR model itself. Finally, the authors also introduce a score for measuring cross-scale consistency.

**Strengths:**

- The authors introduced AID that is a lightweight module for improving the generative performance of VAR models without training/fine-tuning the VAR model itself
- Paper is well written and it was easy to grasp the central idea
- The architectural diagrams were well made and clear
- Performance improvement of 16% in FID for a 3% increase was achieved by the authors.

**Weaknesses:**

Minor:
1. Line 74-74: "Training a disc in contin latent spaces collapses quickly..." - Please provide citation or evidence for this.
2. Please mention if Figure 4 images were cherrypicked or randomly picked in the caption.

Major:
1. A major issue with the paper is that there is no Evidence presented for the central premise of the paper that "once a local mistake enters the chain, subsequent decoders amplify rather than correct it" (Line 39-40) and again on Line 122-124.
-- Is there a citation to support this claim?
-- Is this due to some theoretical deductions? (Unlikely because multilayer nets are universal approximators, and in theory, I don't think there is any reason for why a subsequent decoder cannot correct a mistake in the earlier layers. For instance, LLMs share similar autoregressive nature and seem to be fine).
-- Or is this an empirical claim? Do the authors have evidence to support it by toy experiments or such?
The authors provide one (potentially cherry-picked) qualitative figure in Figure 1 to show how an artifact in the lower layer amplifies to an artifact in the higher scale. But there is no systematic empirical / quantitative experiment to back their claims.
-- What is the Evidence that AID is specifically correcting for such artifacts in lower scales and not affecting non-artifact tokens in the lower scales?

As far as the reader is concerned, the paper presents a small trainable architecture that improves the overall FID performance by some amount. But the central claim is unfounded in the current form of the paper.

2. Second major problem of the paper is the requirement for Cross-Scale Consistency. It is not clear why this is important for a good-quality high res image generation.
Any mistakes in a smaller scale *can* be corrected by layers in the future at a larger scale because these are general universal approximator layers. In such a case, cross-scale consistency actually hinders quality by restricting the amount of correction the layer next can do on the previous scale's noisy tokens.
Infact, with each higher depth in VAR, the model adds more image details that are novel and non-existent in the prev scale. That would reduce the cross-consistency but is not a bad thing.

Moreover, Cross-Scale Consistency must come at the Tradeoff of Diversity/Coverage of Sample Generation, however, no such discussion is presented. If a model has high cross-scale consistency, that should affect the diversity of output samples by reducing it, intuitively.

Cross-Scale consistency may be crucial for applications such as Super Resolution where consistency across scales is necessary but such an application is not presented. I do not see why Cross-Scale consistency is needed for a good quality final high-res generation at the highest scale k, and hence the requirement for the new ISCS score designed by the authors.

3. Section 5.3 - Line 418-422: Figure 4 just has the final high res generations. This is insufficient in my view to make any conclusions on the hypothesis presented in the paper about artifacts in lower scales and AID module fixing that problem. Instead please add a figure that includes image reconstructions at each scale for a given image (picked at random) and compare that with your method. If indeed there exists artifacts in the lower scales for the original VAR method with a worser final high-res output compared to the presented AID method, that would be more convincing.
E.g. the authors talk about a "ghost wing" and yet they don't trace it back to lower scales to show what could have caused it and why the network was unable to correct its mistakes.

4. No Qualitative figures presented in the main paper that show the failure modes of AID module.

5. Section 6: Conclusion - "To address the detail loss and struct distortions in VAR models caused by error accumulation" -- As discussed before, this is an unfounded claim without experimental / theoretical support.

6. If AID-VAR is seen simply as an lightweight trainable module to increase FID of generations without the unfounded claims of cross-consistency being the root of errors, then it must be also compared with similar architectural baselines such as the style injection architecture of StyleGANs using the BatchNorms, among others.

**Questions:**

In addition to the issues raised in Weaknesses,

1. Fake samples are "refreshed" in a GAN training as well, so can the authors please clarify why their on-the-fly disc refresh (Sec 3.3) is novel?
2. In Eq. 8: for a given real image, how are its low-res images computed? What kernel is used to compute low-res? E.g. bicubic kernel or some other anti-aliasing kernel? And, why?

---

### Note · Authors · 2025-11-13

I have read and agree with the venue's withdrawal policy on behalf of myself and my co-authors.